# Axon-Targeting Motifs: Mechanisms and Applications of Enhancing Axonal Localisation of Transmembrane Proteins

**DOI:** 10.3390/cells11060937

**Published:** 2022-03-09

**Authors:** Lloyd J. Steele-Nicholson, Melissa R. Andrews

**Affiliations:** 1Faculty of Environmental and Life Sciences, University of Southampton, Southampton SO17 1BJ, UK; l.j.steele-nicholson@soton.ac.uk; 2Centre for Human Development, Stem Cells and Regeneration, School of Biological Sciences, University of Southampton, Southampton SO17 1BJ, UK

**Keywords:** axon-targeting motif, neuronal polarity, protein trafficking, somatodendritic, transmembrane protein, transcytosis, secretory pathway, axon transport

## Abstract

Neuronal polarity established in developing neurons ensures proper function in the mature nervous system. As functionally distinct cellular compartments, axons and dendrites often require different subsets of proteins to maintain synaptic transmission and overall order. Although neurons in the mature CNS do not regenerate throughout life, their interactions with their extracellular environment are dynamic. The axon remains an overall protected area of the neuron where only certain proteins have access throughout the lifespan of the cell. This is in comparison to the somatodendritic compartment, where although it too has a specialised subset of proteins required for its maintenance, many proteins destined for the axonal compartment must first be trafficked through the former. Recent research has shown that axonal proteins contain specific axon-targeting motifs that permit access to the axonal compartment as well as downstream targeting to the axonal membrane. These motifs target proteins to the axonal compartment by a variety of mechanisms including: promoting segregation into axon-targeted secretory vesicles, increasing interaction with axonal kinesins and enhancing somatodendritic endocytosis. In this review, we will discuss axon-targeting motifs within the context of established neuron trafficking mechanisms. We will also include examples of how these motifs have been applied to target proteins to the axonal compartment to improve both tools for the study of axon biology, and for use as potential therapeutics for axonopathies.

## 1. Introduction

Mature central nervous system neurons display a unique morphology among mammalian cell types. Common multipolar neurons possess a highly polarised morphology consisting of a cell body (soma) with many branching dendrites and a single axon, which may end in a presynaptic terminal at distances ranging from microns to metres away from the cell body, such as in mammals [1]. This polarity of morphology is established de novo early in development, once migrating immature neurons reach their destination within the cerebrum. Immature neurons at first possess many processes but as they develop, one process is specified to become the axon and undergoes rapid outgrowth towards its anatomical target. Axon specification and outgrowth is driven by extracellular cues (e.g., laminin in the extracellular matrix) and intracellular changes such as cytoskeletal remodelling and trafficking of growth machinery into the nascent axon [2].

Having established polarity, mature neurons are considered to consist of two functionally distinct compartments, the somatodendritic and axonal compartments, which are responsible for the reception of post-synaptic potentials, and the initiation and transmission of action potentials, respectively. The distinct complement of proteins localised to either compartment underlies the normal function of mature neurons. Indeed, mislocalisation of axonal proteins may cause or contribute to neuropathology, such as in Alzheimer’s disease and amyotrophic lateral sclerosis [3,4]. Thus, much of the research has focused on characterising the mechanisms by which mature neurons may establish and maintain such polarity of protein localisation between the somatodendritic and axonal compartments, and how these mechanisms change during cellular maturation and go awry in pathology. This review discusses the mechanisms by which neurons maintain polarity of transmembrane protein localisation between the somatodendritic and axonal compartments through protein sorting, trafficking and exclusion mechanisms, and how short peptide axon-targeting motifs (ATMs) interact with these different pathways to promote axonal localisation of transmembrane proteins. Finally, the potential further methodological and therapeutic applications of ATMs are discussed.

## 2. Two Distinct Pathways Mediate Transmembrane Protein Trafficking in CNS Neurons

Transmembrane proteins are initially synthesised and adopt their native confirmation in the endoplasmic reticulum (ER) then undergo further maturation by post-translation modifications as they transit through the Golgi network. In the trans-Golgi network (TGN), transmembrane proteins are segregated into distinct populations of somatodendritic and axonal vesicles, a process that forms the basis of polarity maintenance in neurons. The mechanisms by which axonal cargoes are differentiated from somatodendritic cargoes in the neuronal TGN are not completely understood. However, recent evidence from *Caenorhabditis elegans* suggests that cargo recognition is dependent in part on competition between clathrin-associated adaptor protein (AP) complexes [5]. Binding of AP-3 complexes to axonal cargoes sorts these into vesicles destined for the axon whereas binding of AP-1 targets these to the somatodendritic domain [5,6]. Preferential binding of either AP complex is mediated by dileucine motifs ([D/E]xxxL[L/I]) present in the cytoplasmic tails of certain transmembrane proteins and those with higher affinity for AP-3 thus promoting axonal localisation of transmembrane proteins [5].

Having left the TGN, vesicles containing axonal proteins are trafficked towards both the somatodendritic and axonal compartments, whereas vesicles containing somatodendritic proteins, such as transferrin receptor (TfR), are excluded from the axon [7]. Some axonal transmembrane proteins are trafficked directly to the axon in secretory vesicles without interacting with the somatodendritic membrane, in a pathway resembling the canonical secretory pathway. Such trafficking is known to occur for tropomyosin receptor kinase B (TrkB) [8] and neuron–glia cell adhesion molecule (NgCAM) [9]. One might expect that vesicles carrying axonal proteins are efficiently routed directly to the axonal membrane without interacting with the somatodendritic membrane; however, a second level of transmembrane protein sorting occurs at the somatodendritic surface. Indeed, axonal transmembrane proteins may be exocytosed at the somatodendritic membrane then rapidly reinternalized into endosomes [9,10]. Endosomes containing axonal cargo are then either degraded [9,11] or recycled to the axon in a process resembling transcytosis in epithelial cells [9,12]. Interestingly, binding of ligands to TrkA and type-1 cannabinoid receptor (CB1R) at the axonal or somatodendritic surface, respectively, has been shown to stimulate transcytosis to the axonal compartment [13]. These findings could suggest that the local extracellular environment of both the axon and soma plays a role in regulating the contribution of transcytosis to axonal protein trafficking.

Since finding that these two pathways operate in neurons, their relative contribution to the sorting of axonal proteins has remained somewhat unclear. For example, seminal papers on this topic disagreed on the role of transcytosis in axonal expression of NgCAM [9,12]. With improvement of live-cell imaging of surface labelled proteins this dispute may soon be resolved. Indeed, a recent report using live-cell imaging of hippocampal neurons with surface labelled proteins found that the majority (84–90%) of vesicles containing somatodendritic surface-labelled VAMP2 and NgCAM were targeted to lysosome-like endosomes upon endocytosis and that direct axon-targeting in the secretory pathway accounted for the majority (85–94%) of anterograde vesicles carrying these proteins in the axon [11]. This is a remarkable finding, as it suggests that transcytotic trafficking of axonal proteins is quite inefficient in neurons as much of the axonal transmembrane protein retrieved from the somatodendritic surface is targeted for degradation.

## 3. Maintenance of Transmembrane Protein Polarity by the Proximal Axon

The two previously discussed trafficking pathways for transmembrane proteins in neurons are further supplemented by a host of mechanisms which function to exclude somatodendritic transmembrane proteins from the axonal compartment. Entry of vesicular cargo to the axon is regulated by two distinct regions at the proximal axon, the pre-axonal exclusion zone (PAEZ) and the axon initial segment (AIS).

The PAEZ lies just proximal to the axon hillock within the soma (Figure 1). Within the PAEZ, a population of somatodendritic vesicles are returned to the cell soma without entering the axon [14]. Sorting at the PAEZ appears to be regulated primarily by interactions of cargo-bound kinesins with post-translationally modified microtubules within the PAEZ, since fusion to a kinesin light chain binding sequence or overexpression of an acetylation mimic of α-tubulin results in axonal entry of TfR [14]. Microtubule associated proteins also play a role in filtering cargo at the PAEZ. Microtubule associated protein 2 (MAP2), which localises to the somatodendritic compartment, was shown to inhibit the binding of kinesin-1 to microtubules to favour anterograde trafficking of kinesin-3 bound cargoes in hippocampal neurons and dorsal root ganglion (DRG) neurons [15]. Similarly, the ability of secretory vesicles carrying TrkB to cross the PAEZ was shown to rely on kinesin-3 whereas rapid anterograde trafficking in the axon required kinesin-1 [8]. Thus, even in neurons lacking an AIS, such as DRG neurons [15], entry to the axon and correct distribution of cargo within the axon requires vesicles carrying axonal cargoes to associate with the appropriate complement of kinesins which preferentially bind to and ‘walk’ along axonal microtubules.

Beyond the PAEZ lies the AIS, which makes up the first 20–60 µm of the axon in central nervous system neurons [16]. The AIS is marked by a unique molecular architecture that consists of the cytoskeletal proteins, filamentous actin (f-actin), ankyrin-G (AnkG) and βIV-spectrin that together form an annular cortex or ‘undercoat’ below the plasma membrane of the AIS [17] (Figure 1). Exclusion of somatodendritic transmembrane proteins at the AIS appears to be mediated in part via inhibition of lateral diffusion between the somatodendritic and axonal membranes and a cytoplasmic vesicle filter [18]. AnkG is necessary for the function of the AIS as a selective filter to maintain neuronal polarity since its knockdown allows somatodendritic proteins to cross the AIS and stimulates outgrowth of dendrite-like processes from the proximal axon [19,20]. Interaction with AnkG clusters many transmembrane proteins at high density on the AIS membrane, for example voltage-gated sodium channels (e.g., Nav1.2 and Nav1.6) and cell adhesion molecules (e.g., neurofascin-186 (NF-186) and L1) [21,22,23]. These clustered transmembrane proteins potentially inhibit lateral diffusion of membrane-associated proteins between neuronal compartments by greatly increasing the path length required to diffuse across the AIS due to molecular crowding [24,25].

As previously mentioned, the AIS also represents a cytoplasmic filtering point at which somatodendritic vesicles may be returned to the cell soma [16]. In contrast to somatodendritic vesicles, which pause before undergoing retrograde trafficking at the AIS [7,26], vesicles carrying axonal proteins appear unhindered as they traverse the AIS [22]. Indeed, as the AIS develops it recruits signaling proteins which interact with the cell’s trafficking machinery to exclude somatodendritic cargoes. For example, the ARF6 guanine-exchange factor EFA6 accumulates in the AIS during cortical neuron development where it activates ARF6, which induces retrograde trafficking of vesicles carrying integrins [20,27,28]. Similarly, nuclear distribution element-like 1 (NDEL1) localises to the AIS where it regulates dynein, potentially via lissencephaly 1 (Lis1), to enhance retrograde trafficking of vesicles carrying TfR that enter the AIS [29]. Besides regulation via the activity of microtubule-based motors, actin-based myosin motors also play a role in regulating vesicle trafficking at the AIS. Specifically, myosin Va, a plus-end directed motor, has been shown to be involved in excluding somatodendritic vesicles from the AIS and returning them to the somatodendritic domain, potentially by retrieving these vesicles from microtubules in the AIS where the plus-ends of actin filaments point toward the soma [30]. The accumulation of proteins that promote the activity of retrograde motors associated with somatodendritic cargoes following the initial assembly of the AIS may underlie further development of the AIS as a barrier to axonal entry of proteins. For example, in postnatal day 5 rat pups the AnkG is localized to the AIS however virally-expressed integrins are still permitted entry to the axon whereas in adult animals virally-expressed integrins are completely excluded from the axons of central neurons [31]. This suggests that the mechanical properties of the AIS scaffold alone may not be sufficient to exclude all somatodendritic cargoes and that maintenance of transmembrane protein polarity becomes more stringent with age.

## 4. Axon-Targeting Motifs Exploit Diverse Trafficking Pathways to Promote Axonal Localisation

The aforementioned mechanisms of protein sorting, trafficking and exclusion are dependent on the recognition of cargoes by the neuronal protein sorting machinery. Interactions between cargo proteins and protein sorting machinery are mediated in part by axon-targeting motifs. The term ‘axon-targeting motifs’ is generally used to describe relatively short peptide motifs, usually found in axonal proteins, that when tagged to a non-polarised or somatodendritic protein are sufficient to target it to the axon. Similarly, there also exist mRNA axon-targeting ‘zip-codes’, cis-acting elements that drive axonal localisation of mRNA in the 3′ UTR of axon-enriched mRNAs, for example, in β-actin [32] and the axonal microtubule associated protein tau [33] but these are outside the remit of this review (see review by Turner-Bridge and colleagues, 2020 [34]). ATMs display a variety of targeting mechanisms, reflecting the diversity of trafficking routes that proteins take to reach the axon. This section aims to discuss mechanisms by which ATMs target transmembrane proteins to the axonal compartment (see Table 1 for overview of ATMs discussed in this section).

Several ATMs have been identified that appear to target proteins to secretory vesicles destined for the axon at the TGN. For example, the C-terminus of neurexin-1α (Nxn1α) contains a PDZ-binding motif that targets Nxn1α to axon-targeted vesicles without the protein appearing on the somatodendritic membrane suggesting direct targeting [44]. Beyond interaction with the proteinaceous sorting machinery at the TGN, ATMs from both growth associated protein-43 (GAP-43) and paralemmin contain dicysteine palmitoylation motifs with adjacent dibasic amino acids that anchor these proteins into detergent-insoluble glycolipid-enriched complexes present in the plasma membrane (Figure 2A) [35,38]. These palmitoylation motifs may aid proteins in associating with vesicles in the secretory pathway during TGN sorting, as has been shown for GAP-43 [47]. Indeed, another cytosolic axonal protein, MAP6, also contains a similar N-terminal dicysteine palmitoylation motif with adjacent basic residues (RACCIAR) which promotes association with membranes of Rab6 positive secretory vesicles in the cell soma [48]. These motifs originate in lipid-anchored peripheral membrane proteins and studies using these motifs have tagged them to cytosolic proteins [35,36,49], thus whether they may successfully target transmembrane proteins remains unclear.

As previously discussed, many transmembrane proteins undergo transcytosis to reach the axonal membrane [9,12]. Therefore, increasing endocytosis of a protein from the somatodendritic membrane may increase the pool of endosomes available for transcytosis. Indeed, an early paper by Garrido and colleagues found that fusion of a dileucine-based endocytosis signal from the cytoplasmic domain of Nav1.2 to the non-polarised type 1 transmembrane protein CD4 increased its axonal localisation via clathrin-dependent endocytosis [10]. Another protein which may play a role in selective somatodendritic endocytosis of axonal proteins is the plus-end directed actin-based motor protein myosin VI. By fusing the myosin VI binding domains (MVIBD) of two proteins, optineurin (OPTN) and disabled homologue 2 (DAB2), to the C-terminus of non-polarised CD8 it, was shown that axonal enrichment of this transmembrane protein was dependent in part on endocytosis (Figure 2B) [43]. Further endocytosis-based motifs are also present in mGluR7 and Caspr2 which rely on the interaction with AP-2 and protein kinase C to induce endocytosis, respectively [45,46]. Some endocytosis-based ATMs have also been shown to drive axonal localization by both the transcytotic and secretory pathways, suggesting that sorting mechanisms at the somatodendritic membrane and the TGN may overlap mechanistically. For example, dynasore-mediated inhibition of endocytosis did not completely abrogate axon enrichment of channelrhodopsin tagged with the fused OPTN and DAB2 MVIBDs or the C-terminus Nav1.2 suggesting that these motifs may play a yet uncharacterised role in directing vesicles into the secretory pathway at the TGN [43]. 

Besides the previously discussed mechanisms of sorting proteins into axonal vesicles and removing them from the somatodendritic surface, ATMs may also interact with sorting machinery at the PAEZ and AIS to allow normally somatodendritic cargoes entry to the axon. As previously discussed, vesicles may be permitted entry to the axon and further trafficking to the distal axon by association with kinesin-3 and -1. Indeed, tagging three copies of the kinesin light chain binding sequence TNLEWDDSAI from cargo adaptor protein SifA and kinesin-interacting protein (SKIP) to somatodendritic TfR led to redirection of this protein to the distal axon in hippocampal neurons in vitro [14]. Interaction with axonal kinesin-1 may also underlie the axon-targeting effect of the C-terminal 15 amino acid ATM of amyloid precursor protein (APP) which contains a highly conserved GYENPTY motif [37]. The axon-targeting function of the C-terminus of APP as an ATM was first identified in experiments where fluorescent beads coated with the C-terminus of APP were shown to undergo fast anterograde transport when injected into the giant squid axon suggesting interaction with kinesin-1 [37]. 

Furthermore, the ATM of APP may interact directly with kinesin-1 or indirectly by the binding of jun N-terminal kinase (JNK) interacting protein JIP-1b which in turn interacts with kinesin light chain [50]. Thus, ATMs which drive interaction with axonal kinesins may function by first allowing tagged cargoes to cross the PAEZ and then facilitating fast anterograde transport within the axon itself (Figure 2C).

A subset of axon-targeting motifs containing AnkG binding motifs are found in voltage-gated sodium channels Nav1.2 and Nav1.6 [10,41] and potassium channel Kv3.1 [40] that cluster at the AIS. Nav1.2, for example, contains two putative axon-targeting sequences, an AnkG binding motif present in the second intracellular loop (II-III) and a 9 AA dileucine motif containing ATM in the intracellular C-terminal domain [10]. AnkG binding could increase axonal localisation of transmembrane proteins via two distinct mechanisms. First, lateral diffusion of transmembrane proteins into the AIS from the somatodendritic membrane could allow them to become anchored on the AIS membrane via interaction with AnkG. Secondly, it has previously been shown that Nav1.2 and AnkG are trafficked to the AIS in pre-assembled complexes driven by the interaction of AnkG with kinesin-1 [51] suggesting that AnkG binding motifs may also promote indirect association of transmembrane proteins with kinesin-1 via AnkG (Figure 2D). Whilst these may both be termed ATMs, the pattern of axon-targeting by these motifs appears to differ. By expressing CD4 that was tagged with either loop II-III or the C-terminus of Nav1.2 in hippocampal neurons, Garrido and colleagues showed that loop II-III targets CD4 to the AIS whereas CD4 tagged with the C-terminus was distributed throughout the axon suggesting that extra signals within the C-terminus are required for onward anterograde trafficking [10]. Thus, it appears that AnkG binding is sufficient to increase localisation at the AIS but further ATMs present in these proteins contribute to transport into the distal axon.

## 5. Further Applications of ATMs

### 5.1. Improving Characterisation of Mammalian Neuronal Circuity

Characterising the connectivity of neuronal circuitry is a pre-requisite to fully understanding its function; however, this is made difficult by the complexity of projections within the mammalian brain which may extend over long distances. ATMs could aid such characterisation by enhancing the ability of proteins to enter the axon and accumulate at the distal axon where otherwise they may not readily localise. Indeed, Padmanabhan and colleagues showed that by tagging the red fluorescent protein Tomato with the ATM of APP it was possible to efficiently label axons of the medial forebrain bundle and nigro-striatal tract, improving the ability to count axons within these tracts over non-targeted Tomato [35]. Thus, by simply targeting fluorescent proteins to the axon it could be possible to improve morphological characterisation of neuronal circuitry to study the number, size or complexity of axons.

Another class of fluorescent proteins are the GCaMPs, a family of genetically encoded calcium indicators (GECIs) that increase in fluorescence when Ca^2+^ binds to the protein as a result of conformational changes which exclude water from the fluorophore [52]. Thus, fluorescence can be used as a measurement of transient changes of intracellular Ca^2+^ concentration, such as those that occur during action potential firing [53]. Previous GCaMPs were limited by poor long-distance diffusion of cytosolic GCaMPs to the distal axon, which interferes with precise measurement of Ca^2+^ in the axonal compartment of neurons in vivo [49]. A synapse-targeted GCaMP was produced by fusing GCaMP2 to the cytoplasmic tail of full-length synaptophysin, a synaptic vesicle protein, to monitor spike activity in pre-synaptic boutons of the optic tectum in Zebrafish [54]. Improving on this application, a later study used the dicysteine palmitoylation motif of GAP-43, to develop an axon-targeted version of GCaMP6 for use in mice. This version of GCaMP6 offered lower background signal from somatodendritic domains, improved signal-to-noise ratio and photostability compared to non-targeted GCaMP [49]. 

### 5.2. Improving Genetic Therapies for Axonopathies

Following axotomy, the distal axon displays a stereotypical degeneration process known as “Wallerian degeneration” [55]. The Wallerian degeneration slow (Wld^s^) mouse however displays delayed onset of Wallerian degeneration following axotomy [56]. The Wld^s^ gene encodes a chimeric fusion protein of ubiquitination factor Ube4b and the nuclear NAD^+^ synthesising enzyme nicotinamide nucleotide adenylyltransferase 1 (NMNAT1) [57]. The axoprotective effect of Wld^s^ is dependent on its mislocalisation into the cytoplasm, with a portion of the mutant protein ending up in axons [58]. When NMNAT1 was tagged with the 15AA ATM of APP and its nuclear localisation signal was removed, the axonal localisation and axoprotective properties of NMNAT1 were increased compared to non-targeted NMNAT1 both in vivo and in vitro [36].

Besides potentiating the axoprotective effect of proteins, ATMs may also enhance the pro-regenerative effects of proteins by targeting them to the axon or growth cone. For example, chondroitinase ABC (ChABC), is a bacterial enzyme that degrades the glycosaminoglycan side chains of inhibitory extracellular matrix proteins chondroitin sulfate proteoglycans [59] deposited by reactive astrocytes following spinal cord injury [60]. Infusion of ChABC into spinal cord lesion sites following injury has been shown to promote regeneration of sensory and corticospinal tract axons beyond the lesion site [61]. Several modifications have been made to increase thermostability of ChABC at body temperature [62], allow controllable expression [63], and secretion from mammalian cells [64]. In line with these improvements, a recent in vitro study used the YENPTY ATM from APP to assess the effect of axon-targeting on the pro-regenerative effect of ChABC. Whilst the study did not provide evidence of ChABC being targeted to the neurites or axons of the cells used, potentially due to the lack of antibodies against ChABC, expression of axon-targeted ChABC did increase neurite outgrowth and branching from SH-SY5Y cells and dissociated primary DRG neurons on chondroitin-4-sulfate compared to non-targeted ChABC [65].

Although limited in number these studies suggest that axon-targeting of either axoprotective or pro-regenerative proteins may potentiate their effects by delivering them to the axonal compartment. Several studies have already identified pro-regenerative proteins that are excluded from axons when virally expressed in mature neurons of the sensorimotor cortex and red nucleus, which supply axons to descending motor tracts. These include insulin-like growth factor type 1 receptor (IGF-IR), Trks and integrins [31,66,67]. It could therefore be possible to identify ATMs which could target these proteins to lesioned axons and assess whether this would potentiate the pro-regenerative effects of these proteins in the descending motor tracts. 

## 6. Conclusions

Here we have discussed some of the mechanisms by which neurons establish and maintain a highly polarised distribution of transmembrane proteins between the somatodendritic and axonal compartments. Two main pathways via which transmembrane proteins leave the TGN to reach the axonal membrane have emerged, the indirect transcytosis pathway and the direct secretory pathway. Even for well-defined proteins, the relative contribution of each pathway to axonal localisation has been a point of contention. Recent evidence suggests that the majority of axonal proteins endocytosed from the somatodendritic membrane are trafficked to endolysosomes. Further study of the downstream fate of selectively eliminated axonal proteins will further aid in addressing this problem.

Polarised sorting of transmembrane proteins at the TGN and at the somatodendritic membrane is further coupled to selective filtration at both the PAEZ and the AIS of mature neurons. Evidence from in vitro studies suggests a model of proximal axon vesicle sorting whereby vesicles carrying axonal transmembrane proteins must associate with a specific complement of kinesin motors to traverse both the PAEZ and to gain entry to the AIS where somatodendritic cargoes are physically excluded by the AIS scaffold or by the action of proteins that regulated the motility of somatodendritic cargo-associated retrograde motor proteins. 

Understanding the mechanisms of transmembrane protein trafficking in neurons is necessary to fully characterise how neurons establish and maintain polarity. Further mechanistic characterisation of these pathways will aid in manipulating transmembrane protein trafficking in adult animals. Polarised trafficking may be manipulated both in vitro and in vivo by the addition of ATMs to specific transmembrane proteins. Thus far, many ATMs have been identified and effort has been made to understand the mechanisms by which they are sufficient to induce axonal localisation of normally excluded proteins. It appears clear that ATMs target almost all identified steps within the trafficking and exclusion pathways described here. However, the research on ATMs beyond mechanistic characterisation is relatively sparse. We have described some applications of ATMs but to our knowledge in vivo studies have only been performed on targeting of non-polarised proteins used to study axon biology and potential genetic therapies. Therefore, studies assessing the impact of ATMs on transmembrane receptors localisation in the mature CNS in vivo where mature trafficking and exclusion mechanisms are active are needed.

## Figures and Tables

**Figure 1 cells-11-00937-f001:**
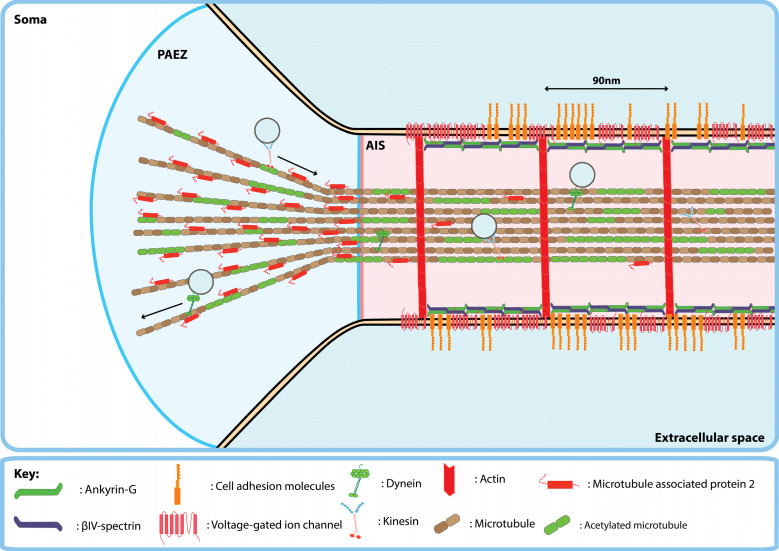
Organisation of the pre-axonal exclusion zone (PAEZ) and axon initial segment (AIS) of mature central nervous system neurons. The PAEZ is located within the soma just proximal to the AIS. In the axon, the majority of microtubules are oriented with the plus-end pointing away from the soma. Plus-end directed kinesins drive anterograde transport and minus-end directed dyneins drive retrograde axonal transport of transmembrane proteins. The AIS is found in the axon hillock and is marked by a unique molecular architecture consisting of periodic rings of f-actin and a submembrane undercoat composed of ankyrin-G and βIV-spectrin. Furthermore, the AIS membrane is densely packed with transmembrane proteins such as voltage-gated ion channels and cell adhesion molecules.

**Figure 2 cells-11-00937-f002:**
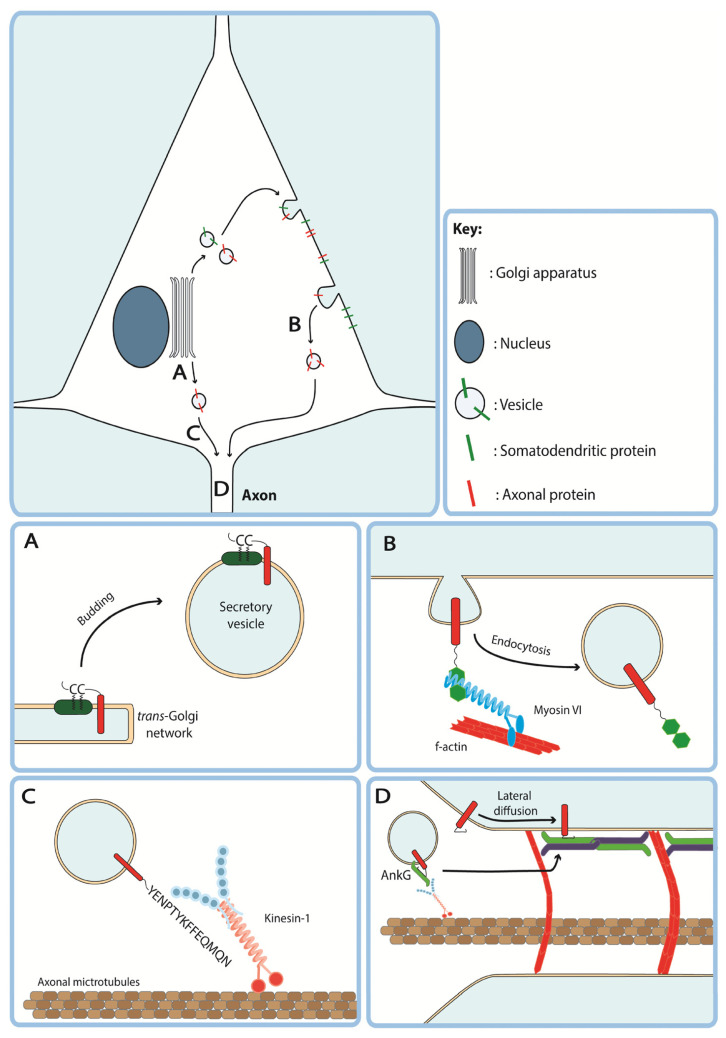
Axon-targeting motifs influence diverse steps in the secretory and transcytotic trafficking pathways of transmembrane proteins to the axonal compartment. (**A**) A dicysteine palmitoylation motif found in Paralemmin and GAP-43 promotes association with lipid rafts in the *trans*-Golgi network, sorting transmembrane proteins into axon-targeted secretory vesicles. (**B**) Fused myosin VI binding domains of optineurin and DAB2 promote endocytosis of transmembrane proteins from the somatodendritic membrane for transcytotic delivery to the axon. (**C**) The C-terminus of APP drives association with light chains of kinesin-1, promoting axonal entry and anterograde trafficking. (**D**) Ankyrin-G (AnkG) binding domains found in Nav1.6 and Kv3.1 promote localisation to the axon initial segment potentially via association with AnkG on kinesin-3 driven vesicles or by anchoring transmembrane proteins in place following lateral diffusion into the AIS.

**Table 1 cells-11-00937-t001:** Summary of previously identified and characterised axon-targeting motifs (ATMs). Examples of several ATMs characterised in the literature. The protein of origin, reported location and amino acid sequence of each ATM is given. Where possible, the putative targeting mechanism is also summarised alongside the model used to study the targeting mechanism.

Protein of Origin	Region	Peptide	Putative Mechanism of Targeting	Model	Source
Amyloid precursor protein (APP)	C-terminus	GYENPTYKFFEQMQN	Promotes interaction with KLC1 and NPTY motifs recruits JIP-1b which interacts with KLC1 to promote association with kinesin-1.	Giant squid axon, primary E18 rat hippocampal neurons, and primary E14-16 murine dorsal root ganglion neurons	[35,36,37]
Paralemmin	C-terminus	DMKKHRCKCCSIM	Dicysteine palmitoylation motif with nearby basic amino acids sufficient for targeting to secretory pathway, likely through association with lipid rafts in trans-Golgi network.	Primary E18 rat hippocampal neurons	[38]
Growth associated protein-43 (GAP-43)	N-terminus	MLCCMRRTKQV			[35,38,39]
Kv3.1	C-terminus	MAKQKLPKKKKHIPRRP	Interacts with T1 tetramerisation domain and Ankyrin-G binding motif.	Primary E18 rat hippocampal neurons	[40]
Nav1.2	C-terminus	CLDILFAFT	Stimulates clathrin-dependent somatodendritic endocytosis.	Primary E18 rat hippocampal neurons	[10]
Voltage gated sodium channel α subunits	Intracellular loop II-III	(V/A)P(I/L)AxxE(S/D)D	Ankyrin-G binding motif.	Primary dorsal root ganglion neuron-Schwann cell myelinating coculture	[41,42]
Optineurin (OPTN)	Myosin VI-binding domains	OPTN AAs 420-526	Association with actin-based minus-end directed myosin VI stimulates somatodendritic endocytosis.	Primary E18 rat cortical neurons	[43]
Disabled homologue 2 (DAB2)		DAB2 AAs 649-719			
Neurexin-1α (Nxn1α)	C-terminus	Nxn-1α AAs 1420-1477	PDZ recognition motif is required for Golgi exit and sorting into secretory vesicles, preferential exocytosis onto axon membrane.	Primary P0 murine hippocampal neurons	[44]
Acetylcholine receptor α4 subunit	M3-M4 loop	[D/E]xxxL[L/I]	AP-2 and -3 binding motif stimulates somatodendritic endocytosis.	Primary P0 rat hippocampal neurons	[45]
Contactin-associated protein-like 2 (Caspr2)	4.1 binding domain	RYMFRHKGT	Protein kinase C phosphorylation of [R/K]X[pS/pT] motif increases somatodendritic endocytosis.	Primary E18 rat hippocampal neurons	[46]
SifA and kinesin-interacting protein (SKIP)	Kinesin light chain binding sequence	TNLEWDDSAI	KLC1 binding motif promotes association with kinesin-1.	Primary E18 rat hippocampal neurons	[14]

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
