# Peer review of "Axon-Targeting Motifs: Mechanisms and Applications of Enhancing Axonal Localisation of Transmembrane Proteins"

_cells, 2022, doi:10.3390/cells11060937_

Round 1

Reviewer 1 Report

This review article provides an overview and discussion of the mechanisms that establish and maintain neuronal polarity and compartmentalization, with an emphasis on the roles of axon-targeting motifs in promoting the specific localization of trans-membrane proteins in axons. This review is very well written and synthesizes important mechanisms that can be leveraged for the study of neuronal function and repair. This is a thus a timely review that I read with a great pleasure and recommend for publication in Cells. The only minor suggestion I would have would be to illustrate the section on the maintenance of transmembrane polarity by the proximal axon with a figure describing the organization and molecular constituents of the PAEZ and AIS for readers who are not versed in axonal biology.

Author Response

We thank the Reviewer for their kind words regarding our manuscript.  We have addressed their one comment below.

1) The only minor suggestion I would have would be to illustrate the section on the maintenance of transmembrane polarity by the proximal axon with a figure describing the organization and molecular constituents of the PAEZ and AIS for readers who are not versed in axonal biology.

Response: A new figure (figure 1) was inserted on page 4 to show the molecular architecture of the PAEZ and AIS in line with what was previously described in the reviewed text. This figure is found on page 4 and referenced in the text on lines 125 and 145. (The previous Figure 1 is now Figure 2).

Reviewer 2 Report

This review provides a balanced view and is very well written, a welcome and timely piece, indeed.

Author Response

We thank the reviewer for the kind comments about our manuscript.

Reviewer 3 Report

The manuscript from Steele-Nicolson and Andrews titled “Axon-Targeting Motifs: Mechanisms and Applications of Enhancing Axonal Localisation of Transmembrane Proteins” is a short review describing the current knowledge regarding the mechanisms controlling the sorting and axonal localization of transmembrane proteins. In particular, the review concentrates on the role of axon-targeting motives in regulating the access and further targeting of transmembrane proteins to the axon. Finally, the review introduces a couple of possible application for the axon-targeting motives to target transmembrane proteins with the aim of addressing their roles in axonal biology or in the therapy of axopathologies.

The topic is relevant in the field of neurobiology and the input about using axon-targeting motifs as tool in research or therapy is of interest. The review is clearly written and provides a comprehensive picture of the current knowledge on this topic. I think that this review should be accepted for publication. I only have a couple of minor comments that should be addressed before publication:

  • I find confusing that Figure 1C is described before figure 1B. What is the reason for this? Can this not be changed according to the alphabet?
  • “In to” at line 191 should be changed to “into”
  • The phase “….dynasore-mediated blockade based? endocytosis did not completely…” at line 220 should be corrected
  • The phrase “…..in experiments where fluorescent beads decorated with the c-235 terminus of APP where it was shown to undergo fast anterograde transport when…” at line 235 should be corrected.

Author Response

We thank the Reviewer for their kind words about our manuscript.  We have addressed their concerns below.

1) “I find confusing that Figure 1C is described before figure 1B. What is the reason for this? Can this not be changed according to the alphabet?”

Response:  Panels B & C in figure 1 (now figure 2) were swapped and the references in the text were changed (lines 251 and 469) so that the references in the text appear in alphabetic order.

2) ““In to” at line 191 should be changed to “into”.”

Response: The sentence now reads “…that anchor these proteins into detergent insoluble…” (line 230).

3) “The phase “….dynasore-mediated blockade based? endocytosis did not completely…” at line 220 should be corrected.”

Response: The sentence now reads “…dynasore-mediated inhibition of endocytosis did not completely” (line 256)

4) “The phrase “…..in experiments where fluorescent beads decorated with the c-235 terminus of APP where it was shown to undergo fast anterograde transport when…” at line 235 should be corrected.”

Response: the sentence now reads “in experiments where fluorescent beads coated with the C-terminus of APP were shown…” (line 457).